# An Effective Prophylactic and Therapeutic Protection Against Botulinum Type A Intoxication in Mice and Rabbits Using a Humanized Monoclonal Antibody

**DOI:** 10.3390/toxins17030138

**Published:** 2025-03-14

**Authors:** Chi Ho Yu, Young-Jo Song, Dong Hyun Song, Hae Eun Joe, Chang-Hwan Kim, Hyungseok Yun, Na Young Kim, Euni Sim, Seong Tae Jeong, Gyeung Haeng Hur

**Affiliations:** 1Agency for Defense Development, Yuseong, P.O. Box 35, Daejeon 305-600, Republic of Korea; ch675@add.re.kr (C.H.Y.); swpia@add.re.kr (D.H.S.); jhaeeun@add.re.kr (H.E.J.); vetkim@add.re.kr (C.-H.K.); birdyjeong@add.re.kr (S.T.J.); 2ABION Inc., Seoul 08394, Republic of Korea; elf4040@abionbio.com (N.Y.K.); euni@abionbio.com (E.S.)

**Keywords:** botulinum neurotoxin, monoclonal antibody, mouse, rabbit

## Abstract

Botulinum neurotoxins (BoNTs) are the most potent toxins on Earth and are classified as Category A biological agents. BoNTs lead to paralysis in humans and cause botulism. Antibody therapeutics can effectively treat toxin-mediated infectious diseases. In this study, we generated a pharmaceutical humanized monoclonal antibody (HZ45 mAb) to prevent or treat botulism. HZ45 binds to the heavy chain receptor (HCR) domain of the toxin, preventing the toxin from entering the cell. The mAb was produced using hybridoma technology and phage display. We evaluated HZ45 mAb for the neutralization of BoNT serotype A (BoNT/A) in mice and rabbits. The survival results showed that pretreatment with HZ45 mAb provided 100% protection at a dose of 0.1 mg per mouse against a maximum of 100 LD_50_ of BoNT/A. To assess the therapeutic efficacy of HZ45 mAb in New Zealand white rabbits (NZWs), a 5 mg dose was administered 4 or 8 h after challenge with 10 LD_50_. The results indicated that 5 mg of HZ45 could treat the NZWs within 8 h after exposure to 10 LD_50_ botulinum. Consequently, in an in vivo context, including mice and rabbits, HZ45 mAb could protect against botulinum type A intoxication.

## 1. Introduction

Botulinum neurotoxins (BoNTs) are the most potent toxins produced by the anaerobic Gram-positive spore-forming bacterium *Clostridium botulinum* [1]. These toxins exist in eight botulism neurotoxin serotypes: A, B, C, D, E, F, G, and H [2,3]. Serotypes A, B, E, F, and H cause a serious human disease called botulism, which can naturally occur through ingestion of the toxins in food (foodborne botulism) or infection of wound sites (wound botulism). In particular, infant botulism occurs through foodborne diseases and can pose a public health threat [4,5,6]. Theoretically, a 70 kg person can be killed by inhaling 0.7 to 0.9 µg, or by ingesting 70 µg [7]. The clinical symptoms constitute acute flaccid paralysis involving the muscles of the eyes, face, and pharynx by blocking the release of acetylcholine from synapses at the neuromuscular junction. As the disease progresses, patients die due to respiratory failure caused by paralysis of the pharynx, diaphragm, and intercostal muscles [4,8].

Current therapies for botulism include intensive care and antitoxin treatment. Two antitoxins have been licensed to treat botulism. One is equine antitoxin (BAT)^®^ [9] and the other is human botulism immune globulin (BIG-IV) [10]. Equine antitoxin has a short half-life (7.5–34.2 h) and can cause serum sickness in the human immune system. Large-scale manufacture of BIG-IV is not easy compared to producing mAb using a cell line; therefore, the limited quantity precludes its application for large-scale passive immunization of adults [10].

Recently, the development of human or humanized monoclonal antibodies (mAbs) that can be produced on a large scale and at excellent quality has been introduced as a medical countermeasure [11,12]. The mAbs’ well-defined specificity could lessen their toxicity relative to immunized serum, and human or humanized mAbs can treat botulism with fewer side effects compared to equine antitoxin. The potency of mAb combinations can be increased by two to three orders of magnitude compared to the use of single mAb [13,14]. At least three antibodies that bind non-overlapping epitopes are necessary for effective protection. Therefore, we need to secure more single mAbs with good efficacy. However, securing a single mAb takes a lot of time and money. Thus, there is an urgent need to produce a highly effective single neutralizing antibody targeting botulism. In this study, we aimed to develop a highly potent humanized mAb. We generated a humanized monoclonal antibody for BoNT/A using phage display library technology. The binding affinity and neutralizing efficacy were evaluated through in vitro and in vivo models to identify targets for the design of antibody therapeutics. The selected mAb was engineered to have a distinct humanized variable region that binds the heavy-chain constant region of BoNT/A. Conclusively, we found that this single humanized mAb protected mice and rabbits against BoNT/A.

## 2. Results

### 2.1. Selection of scFv Clone and Full-Length Humanized Antibody

Using the purified HCBD as an antigen, bio-panning was performed. The preparation was subjected to three rounds of panning. We initially selected 529 clones and 105 clones in the second round. Of 105 clones, 12 clones were selected by means of ELISA in the 3rd round of panning. We selected clones with OD values greater than 0.5. Each soluble scFv was expressed from 12 clones, and the affinity with BoNT/A was confirmed by ELISA. According to the ELISA results (Figure 1A), 3 out of 12 clones were converted to full-length antibodies. The final three full-length antibodies were selected through antigen-binding ELISA (Figure 1B). Among the three full-length antibodies, one final antibody (HZ45 antibody) was selected through murine bioassay. The amino acid sequence of the HZ45 antibody was confirmed (Figure 2).

### 2.2. Binding Affinity

The measured kinetic dissociation constant (K_D_, 95% confidence interval) of the antibody against the holotoxin was 5.513 × 10^−10^ M; it exhibited 10-fold better affinity (4.23 × 10^−11^ M) against the C-terminal domain of the neurotoxin (Figure 3). Owing to the observed affinity, we found that Hz45 has excellent affinity against BoNT/A, especially for the C-terminal domain. Since that domain is responsible for the cellular recognition of the toxin for transmembrane transport, the neutralizing ability of this antibody might come from inhibiting the cellular uptake of the toxin.

### 2.3. SNAP-25 Cleavage Assay with HZ45

Cell-based toxin neutralization efficacy was analyzed using the SNAP-25 cleavage assay. The prevention of cleavage of the SNAP-25 protein was detected in the Neuro-2A cells with the HZ45 antibody mixture. Cleavage of the SNAP-25 protein decreased in an antibody-concentration-dependent manner (Figure 4). A 1 µg/mL concentration of HZ45 antibody showed toxin neutralization efficacy.

### 2.4. In Vivo Neutralizing Assay with HZ45

In vivo toxin neutralization was performed using a murine bioassay. At first, 1000 µg of HZ45 protected mice challenged with 1000 LD_50_. None of the mice displayed botulism-specific symptoms such as wasp waist during the entire 28-day experiment. Next, we administered lower amounts of the antibody. The mice that received 100 µg of HZ45 survived against BoNT/A at 100 LD_50_. However, the mice challenged with BoNT/A at 100 LD_50_ received no protection from 50 µg of HZ45. We then tested the effective dose of HZ45 against BoNT/A at 20 LD_50_. The mice that received 5, 10, or 50 µg of HZ45 survived against BoNT/A at 20 LD_50_, but 1 µg of HZ45 did not protect the mice challenged with BoNT/A at 20 LD_50_ (Table 1).

### 2.5. Protection of Mice Against BoNT/A Intoxication

The prophylactic efficacy results showed that pretreatment with HZ45 provided significant protection at a dose of 100 μg per mouse against 100 LD_50_ of BoNT/A. To determine the minimal effective dose of HZ45, a group of mice was pretreated via intravenous injection with a lower dose of HZ45 (50 or 10 μg per mouse) and challenged 24 h later with 20 LD_50_. The administration of HZ45 at a dose of 50 μg per mouse resulted in significant protection against BoNT/A at 20 LD_50_ (Table 2).

To assess the therapeutic efficacy of HZ45, a 100 µg dose was administered intravenously to mice after challenge with 20 LD_50_. As shown in Table 2, when the treatment was administered 8 h after toxin administration, all the mice survived. We tested to confirm the therapeutic effect against 50 LD_50_, but when 50 LD_50_ was administered, the mice were moribund at around 8 h.

### 2.6. Protection of Rabbits Against BoNT/A Intoxication

To assess the prophylactic efficacy of HZ45, the NZWs were pretreated with HZ45 at a dose of 5 mg/rabbit via intravenous administration, and they were challenged 24 h later with 10 LD_50_. The survival results showed that the NZWs administered with HZ45 were protected completely against the challenge with 10 LD_50_ of BoNT/A (Table 3).

To assess the therapeutic efficacy of HZ45 in the NZWs, a 5 mg dose was administered to the NZWs 4 or 8 h after challenge with 10 LD_50_. No specific clinical symptoms were observed in rabbits administered the treatment. The results are summarized in Table 3. These results indicate that 5 mg of HZ45 could treat the NZWs within 8 h after exposure to 10 LD_50_ of BoNT/A. We performed experiments to determine the therapeutic effect against 100 LD_50_, but when 100 LD_50_ was administered, the rabbits were moribund at around 8 h.

## 3. Discussion

BoNTs are categorized as Category A biological agents and recognized as potential biological weapons due to their exceptional toxicity and ease of production [4]. Currently, the treatment for BoNT intoxication is antiserum [15]. Antisera are difficult to acquire because they require the immunization of horses or humans. In addition, each produced antiserum has a different antibody composition and potency. To solve these problems, we developed a humanized monoclonal antibody for the treatment of botulism. At first, hybridomas were obtained and screened for the selection of anti-BoNT/A murine mAbs, using an indirect ELISA with the heavy-chain binding domain (HCBD) of BoNT/A. Several positive hybridomas were selected, and a murine bioassay was performed against 20 LD_50_ of BoNT/A.

To analyze the kinetic interaction between the HZ45 antibody and holotoxin or HCBD, we employed surface plasmon resonance technology. The Kd value was determined from the increase rate of the Biacore signal during binding and the decrease rate during the wash-off interval. Figure 3 shows examples of the binding kinetics between HZ45 and BoNT/A or HCBD. The resulting HZ45 bound to the HCBD with an apparent Kd of 0.042 nM—an affinity 10-fold higher (i.e., lower Kd) than that of BoNT/A. Previous studies have examined Kd values between mAbs and HCBDs or holotoxins. Pless et al. cloned eleven neutralizing mAbs, and their overall Kd values ranged from ~0.9 to 0.06 nM [16]. Razai et al. generated two neutralizing ScFv antibodies, and their Kd values ranged from picomolar to sub-picomolar (1.71 and 0.554 × 10pM [17]. Miethe et al. developed neutralizing scFv-Fc against the BoNT/A light chain from a macaque immune library; they selected four final candidates mAbs, and their affinities were in the sub-nanomolar range [18].

Next, in vivo toxin neutralization was performed using a murine bioassay in which BoNT/A and the antibody were premixed and injected intraperitoneally. Murine bioassays have generally been used for several decades to analyze the neutralization ability of BoNTs [16,19,20]. Only 10 μg of HZ45 was able to completely protect mice challenged with 20 LD_50_, while 1 µg of the mAb failed to protect mice from 20 LD_50_. This result demonstrates that 10 µg of humanized mAb was effective against BoNT/A at 20 LD_50_. Previous studies have that shown the majority of monoclonal antibodies protect against a toxin at 20 LD_50_ [16,21,22]. Pless et al. showed that a murine mAb protected mice against 10 LD_50_ of BoNT/A, while Yu et al. found that no single mAb completely protected mice from 20 LD_50_ of BoNT/A (their mAb merely prolonged the time to death when challenged with 20 LD_50_). Nowakowski et al. showed that 50 µg of mAb prolonged the time to death but failed to protect mice from 20 LD_50_ of BoNT/A. Adekar et al. showed that 100 µg of mAb completely protected mice from doses of up to 25 LD_50_ of BoNT/A [23].

Most previous studies have shown prophylactic protection from mAbs. Here, we found that HZ45 provided therapeutic protection when administered 8 h after challenge with the toxin. In the mice, significant protection was seen with the HZ45 mAb administered 8 h after a 10 LD_50_ challenge with BoNT/A. However, the PBS control group died at 20–24 h against 20 LD_50_ of BoNT/A. These results demonstrate that HZ45 could neutralize BoNT/A after botulinum intoxication in the mice. In the rabbits, significant protection was observed with the HZ45 mAb administered 8 h after a 10 LD_50_ challenge with BoNT/A. However, the PBS control group died at 24–28 h against 10 LD_50_ of BoNT/A. These results demonstrate that HZ45 could neutralize BoNT/A after botulinum intoxication in rabbits.

Recently, other studies have shown that a combination of several mAbs may synergistically neutralize BoNT [14,21,22,23,24,25]. Yu et al. screened two IgGs (1B6 and C10) from a fully synthetic human single-chain variable fragment library. When this combination of antibodies was used, the combined mAbs completely neutralized 1000 LD_50_ of BoNT/A. Adekar et al. developed two IgGs, 6 A and 4LCA, which were generated using hybridoma technology from humans immunized against pentavalent botulinum toxoid. Fifty micrograms of the mAbs was premixed with 1000 LD_50_ of BoNT/A, and the survival rate was 100%. Nowakowski et al. generated mAbs via phage display; although no single mAb significantly protected the mice, a combination of three mAbs completely protected the mice against 45,000 LD_50_ of BoNT/A—a potency 90 times greater than that of human immunoglobulin. Garcia-Rodriguez et al. showed that three monoclonal antibodies underwent molecular evolution in order to enhance their affinity in the BoNT/C and D. Fan et al. demonstrated that an equimolar mixture of the mAbs effectively neutralized BoNT/F1, F2, F4, and F7 in the mouse neutralization assay.

Currently, the most advanced mAbs for the treatment of botulinum are those of XOMA 3AB, consisting of three IgG1 monoclonal antibodies in a combined therapeutic agent against BoNT/A botulism that has completed phase 1 clinical trials [11]. XOMA 3AB is an equimolar mixture of three IgG1 mAbs that possess different epitopes of BoNT/A. Each mAb was generated to have distinct human or humanized variable regions and human light- and heavy-chain constant regions. 

## 4. Conclusions

In this study, prophylactic and therapeutic results demonstrated that the HZ45 antibody could neutralize BoNT/A within the mice and rabbits’ blood circulation. Therefore, the HZ45 antibody shows promise for the treatment of BoNT/A as a therapeutic and prophylactic agent. In follow-up studies, nonclinical studies, including pharmacokinetics, pharmacodynamics, and toxicology of HZ45, will be performed. For non-clinical studies, we require the mass production of antibodies and intend to investigate the development of cell lines with high antibody expression.

## 5. Materials and Methods

The experimental design is summarized in Figure 5.

### 5.1. Botulinum Toxin Purification

*Clostridium botulinum* serotype A (subtype A1) was cultured for 4 days in trypticase-pepton glucose yeast extract (TPGY) broth medium (Kisan Bio, Seoul, South Korea) under anaerobic conditions. The cultured medium was harvested via centrifugation, and the progenitor toxin complex was precipitated by the addition of sulfuric acid. The precipitants were collected via centrifugation and resuspended in 0.05 M PBS buffer at pH 7.4. After two hours of stirring, it was re-centrifuged to remove precipitating impurities. The supernatant was then stirred with continuous addition of ammonium sulfate up to a 60% *w*/*v* ratio. Precipitated toxins were collected via centrifugation and resuspended in the same buffer used above. Dialysis was performed to remove any remaining ammonium sulfate from the solution. The sample was then loaded onto an ion-exchange chromatograph (HiTrap DEAE, GE Healthcare Life Science, Chicago, IL, USA). The flow-through was collected, concentrated via ultrafiltration (Merck-Millipore, Billerica, MA, USA), and loaded onto a size-exclusion chromatograph (Superdex 200, GE Healthcare Life Science, Chicago, IL, USA). Endotoxins were removed using endotoxin affinity columns and anion exchange chromatography. The toxicity of the purified product was evaluated via a lethal dose assay on ICR mice in our previous study [26].

### 5.2. Hybridoma Assay

#### 5.2.1. Antigen Preparation

The recombinant C-terminal domain and toxoid of BoNT/A were used as antigens. The recombinant C-terminal domain of BoNT/A was generated as described in a previous study [24]. Briefly, to prepare the recombinant domain, the BoNT/A-heavy chain receptor (HCR) gene was cloned into pGEX-4T-1 (GE Healthcare Life Science, Chicago, IL, USA) and digested with *BamHI* and *XhoI* (New England Biolabs, Ipswich, MA, USA). The cloned plasmid was then transformed into *E. coli* BL21(DE3), and the expressed protein was purified via affinity and size-exclusion chromatography. To prepare the BoNT/A toxoid, purified BoNT/A was dialyzed in 100 mM PBS containing 0.4% formaldehyde at 35 °C for 7 days.

#### 5.2.2. Immunization

The female BALB/c mice (Orient Bio, Sung-nam, South Korea) were first immunized using the prepared BoNT/A HCR and toxoid with complete Freund’s adjuvant (Sigma-Aldrich, St Louis, MO, USA) via peritoneal injection. Complete Freund’s adjuvant was used for the first immunization, and incomplete Freund’s adjuvant was used for the following injections at weeks 2 and 4.

#### 5.2.3. Hybridoma Cell Generation and Indirect ELISA

Cell fusion to generate hybridoma cells was carried out, as described by Kohler et al. (1976) [27]. The mouse with the highest IgG titer at three days after final immunization was euthanized with CO_2_, and its spleen was dissected. Then, 10^8^ viable spleen cells and 10^7^ viable SP2/0-Ag14 myeloma cells were fused by slowly mixing them in 50% PEG 4000 (Sigma-Aldrich, St Louis, MO, USA). The fused cells were cultured in HAT medium on a 96-well plate. Cells producing antibodies were measured via indirect ELISA using 0.2 ug/mL of HCBD as antigens. The HCBD was produced in a previous study [25]. Antigen coating was performed for 12 h at 4 °C, followed by 2 h of blocking with 2% skimmed milk at 37 °C. Antibody-containing sera were then treated for 2 h before being blocked, as previously described, and a 1/2000 dilution of horseradish peroxidase (HRP)-conjugated goat anti-mouse IgG was treated as a secondary antibody. Hydrogen peroxidase and o-phenylenediamine (OPD) were used for visualization. Washing with phosphate-buffered saline with 0.05% Tween 20 (PBST) was performed 3 times between treatments. The OD450 was measured.

### 5.3. Sequence Analysis of the Murine BoNT/A Antibody

Of these, one hybridoma that produced the toxin-specific neutralizing monoclonal antibody from immunized mice was screened out. The heavy-chain and light-chain syntheses for the humanized scFv library were carried out by means of overlap PCR using the murine monoclonal antibody as a template. Instead of choosing the most homologous germ line as the framework for library production, we used the Herceptin framework (IGKV1, IGHV3), which is widely used in clinical practice. It is known that this framework has low immunogenicity and is structurally very stable [28,29]. The complementarity-determining regions (CDRs) were selected by means of Kabat numbering. The *E. coli* containing these libraries were cultured, and a library pool was constructed using phage display. The diversity of the final designed humanized scFv library was 7.55 × 10^7^ (LC diversity: 6144, HC: 1.23 × 10^4^).

The variable regions of the heavy chain and light chain of the antibody were amplified using RNA obtained from hybridoma cells. A murine scFv primer mixture (Pharmacia Biotech, Uppsala, Sweden) was used. Briefly, PCR was performed after mixing 10 μL of 5× buffer, 2 μL of dNTP mix, 1 uL of primer mix, 50 ng of RNA, and 2 uL of enzyme mix. The PCR product was cloned using the pGEM-T vector system protocol (Promega, Madison, WI, USA). After cloning and transformation, the colonies obtained were cultured, the DNA was isolated, and the sequence was analyzed.

### 5.4. scFv Library Construction

The Herceptin antibody gene was used as a framework for the construction of a library of humanized scFv using a mouse antibody template. Heavy-chain and light-chain syntheses for the scFv library construction were performed via overlap PCR (Appendix A). The heavy and light chain’s design strategy is as follows. Briefly, among the framework sequences, the places where the mouse sequence was important were fixed to the mouse sequence [30]. Deamidation site 96D97G was found in HCDR3, and randomization of 97G was performed. The NNS codon was inserted into the hot spot site of somatic mutation (present in CDR2). No deamidation site was found in the light chain. The NNK codon was inserted into the hot spot site of somatic mutation (present in CDR3). For the heavy chain, PCR was performed using a total of 24 overlapping primers from 45VHS1-1 to 45VHS1-24. For the light chain, PCR was performed using a total of 20 overlapping primers, from 45L1-1 to 45L-20. PCR was performed using speed pfu polymerase, and after PCR was completed, the pCmb3xss plasmid vector and PCR product were digested with the SfiI restriction enzyme. After that, ligation was performed to prepare the phagemid library, and they were transformed into the *E. coli* SS320 strain to obtain a library. *E. coli* SS320 was infected with helper phage and isolated by PEG precipitation. And then, isolated library phage was used by bio panning and phage display.

### 5.5. Bio-Panning and Selection of scFv Clone

To perform bio-panning, *E. coli* containing the library was cultured, and the phage pool was created by infecting it with the M13KO7 helper phage. BoNT/A antigen-coated immunotubes were used for the primary panning of the produced phage pool. The immunotubes were coated with 5 μg/mL of antigen and blocked with 4% skimmed milk for 2 h at 37 °C. The phage and skimmed milk were mixed at a 1:1 ratio and incubated in the immunotubes for 2 h. The bound phage was eluted with 100 mM triethylamine, neutralized with 2 M Tris pH 8.0, and infected with XL1 blue. After incubation at 37 °C for 30 min, the phage was centrifuged (3500 rpm, 10 min) and spread on a SOBAG medium plate after resuspension in 2 × YT medium containing 2% glucose. After the 2nd round of bio-panning, antigen-specific clones were selected via phage ELISA.

### 5.6. Expression of Soluble scFv

The expression method for soluble scFv is as follows. The selected clones were inoculated into 1 mL of 2YT/CAR/TET medium and cultured overnight at 37 °C. The culture was diluted 1/100, inoculated into 2YT/CAR/TET medium, and cultured. 1 mM IPTG was added to the culture medium and cultured overnight at 25 °C to prepare soluble scFv supernatant. ELISA was performed using the supernatant to select candidate clones.

### 5.7. Full-Length Humanized IgG Antibody Conversion of scFv and Sequence Analysis

Selected clones in the ELISA were converted to full-length antibodies. Overlapping PCR was performed to amplify the heavy and light chains. In the heavy chain, the pCEP5 plasmid was digested with Bgl II/Age I. In the light chain, the pCEP5 plasmid was digested with Bgl II/Bsiw I. The PCR product was inserted using an Ez-fusion cloning kit. The full-length humanized antibody sequence was analyzed. ELISA of the full-length antibodies was performed to select the final antibody.

### 5.8. Expression and Purification of the Humanized Antibody

After the heavy chain and light chain of the final clone were codon-optimized [29], they were cloned into an expression vector (p458-HZ45-2-heavy-OPT-DHFR and p459-HZ45-2-light-OPT-GS) and treated with *BamHI* restriction enzyme to form a linear plasmid before being transfected into CHO-K1 (ATCC no.: CCL-61) cells. The cells were seeded at 2.5 × 10^6^ cells/plate, and 15 μg of plasmid, 30 μL of Lipofectamine 3000 (Thermo Fisher, Waltham, MA, USA), and 500 μL of Opti-MEM (Thermo Fisher, Waltham, MA, USA) mixture were used for transfection. The transfected cells were seeded and cultured at 3 × 10^4^ cells/well after 48 h. The cell culture medium was 50 uM MSX, 1 × GS supplement, 10% FBS, and IMDM (Thermo Fisher, Waltham, MA, USA).

The produced antibody was harvested by centrifuging the culture medium. Then, ammonium sulfate was added to the supernatant gradually until the concentration reached 30% *w*/*v*. After 5 h of stirring under refrigerated conditions, the mixture was centrifuged for 20 min at 8000× *g*. Ammonium sulfate was added again to the supernatant, reaching a final concentration of 40% *w*/*v*. Then, the same stirring and centrifugation were performed; however, the pellet was collected at this time. After the pellet was resuspended in a buffer containing 20 mM sodium phosphate (pH 7.0), dialysis was performed using the same buffer. The supernatant from the consequent centrifugation was added to ammonium sulfate until a final concentration of 1 M. The sample was then loaded onto an HIC-Phenyl column (GE Healthcare Life Science, Chicago, IL, USA), equilibrated with 20 mM sodium phosphate (pH 7.0) and 1 M ammonium sulfate, and eluted with a gradient generated with buffer free of ammonium sulfate. Fractions containing antibodies were selected via UV absorbance and SDS-PAGE.

### 5.9. Binding Affinity

The produced HZ45 antibody was tested for in vitro affinity against the designed antigen, botulinum type A neurotoxin. The affinity of the purified antibody against the responsible antigens was measured using surface plasmon resonance (Biacore, GE Healthcare Life Science, Chicago, IL, USA). Here, 50 ug/mL of HZ45 in HBS-EP buffer (10 mM HEPES pH 7.4, 0.15 M NaCl, 3 mM EDTA, 0.005% *v*/*v* surfactant P-20) was immobilized on a CM5 chip. Botulinum type A neurotoxin and the C-terminal domain of its heavy chain (AHC) were used as antigens for the measurement. Each antigen was prepared in an HBS-EP buffer at concentrations ranging from 0 to 64 nM by means of twofold serial dilutions. After 120 s of flow of the diluents, dissociation between the antibody and each antigen was monitored for 600 s.

### 5.10. SNAP-25 Cleavage Assay

The cell-based neutralization efficacy of the HZ45 antibody was evaluated via the SNAP-25 cleavage assay [31]. Neuro-2a cells were seeded in a 24-well plate with MEM and treated with ganglioside for 24 h. After the ganglioside treatment, the cells were treated with 5 ug/mL botulinum toxin, HZ45 antibody, and anti-SNAP-25 antibody (Thermo Fisher, Waltham, MA, USA) at varying concentrations for 2 days. The cells were then harvested, and Western blot analysis was performed to confirm SNAP-25 protein cleavage.

### 5.11. Animal Studies

Female 6-week-old ICR mice (Orient Bio, Sung-nam, South Korea) were obtained from Orient Bio. The mice were allowed at least 7 days to acclimate before administration. A murine bioassay was performed using female ICR mice. Briefly, the HZ45 antibody was premixed with a range of murine LD 50 BoNT/A in a total volume of 0.5 mL of gelatin phosphate buffer and incubated at room temperature for 1 h. The mixture was then injected intraperitoneally (Table 1). Next, prophylactic and therapeutic efficacy tests were performed. The mice were injected intravenously with various amounts of HZ45 antibody at various timepoints, as indicated in Table 2. For challenge with BoNT/A, the mice were injected intramuscularly.

To confirm the efficacy in different animals, a rabbit study was conducted using New Zealand white rabbits (NZWs). Female 2–3 kg New Zealand white rabbits were obtained from Duyeol Biotech (Seoul, South Korea). The rabbits were allowed at least 14 days to acclimate before the experiment. To assess the prophylactic and therapeutic efficacy, the rabbits were injected intravenously with various amounts of HZ45 antibody at various timepoints, as indicated in Table 3. The control animals were treated with an irrelevant human IgG mAb (Merck Life Science, St. Louis, MO, USA). For challenge with BoNT/A, the animals were injected intramuscularly. Survival and clinical signs were monitored until 28 days after the toxin injection. The Animal Care and Use Committee at the Agency for *** Development approved the mouse and rabbit experiments (ADD-IACUC-22-21).

## 6. Patents

The amino acid sequences of the antibody obtained in this study were registered in a Korean patent (Patent number 10-1958312).

## Figures and Tables

**Figure 1 toxins-17-00138-f001:**
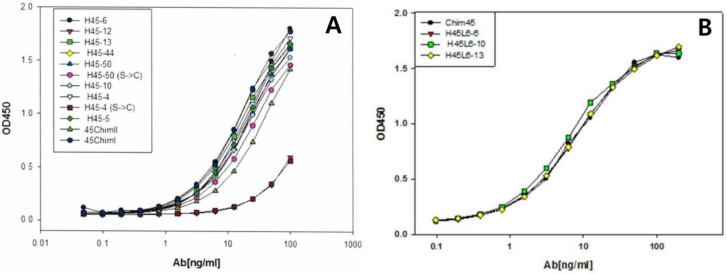
ELISA results. (**A**) Binding of 12 soluble scFvs with BoNT/A was analyzed using ELISA. (**B**) Three full-length humanized antibodies with BoNT/A was analyzed using ELISA.

**Figure 2 toxins-17-00138-f002:**
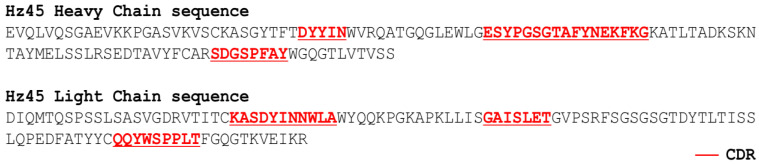
The light-chain and heavy-chain amino acid sequences of HZ45.

**Figure 3 toxins-17-00138-f003:**
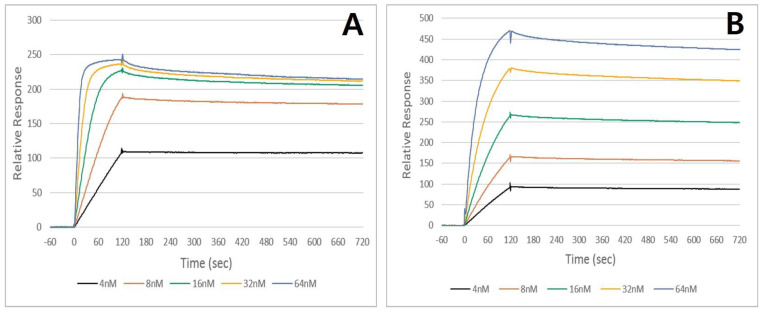
Sensorgram for measuring the dissociation constant between (**A**) Hz45 and the C-terminal domain of BoNT/A and between (**B**) Hz45 and BoNT/A.

**Figure 4 toxins-17-00138-f004:**
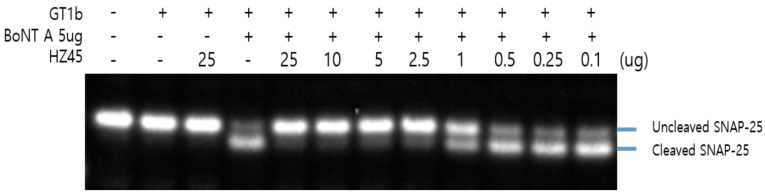
Protection of protein by the HZ45 antibody. The Neuro-2a cells were exposed to ganglioside for 24 h. The cells were treated with the indicated dose of BoNT/A and HZ45 antibody. In lane 2, irrelevant IgG was treated. After two days of treatment, the cells were collected and lysed. Lysates were subjected to Western blotting using anti-SNAP-25 antibodies. Cleavage of the SNAP-25 protein was inhibited by the HZ45 antibody.

**Figure 5 toxins-17-00138-f005:**
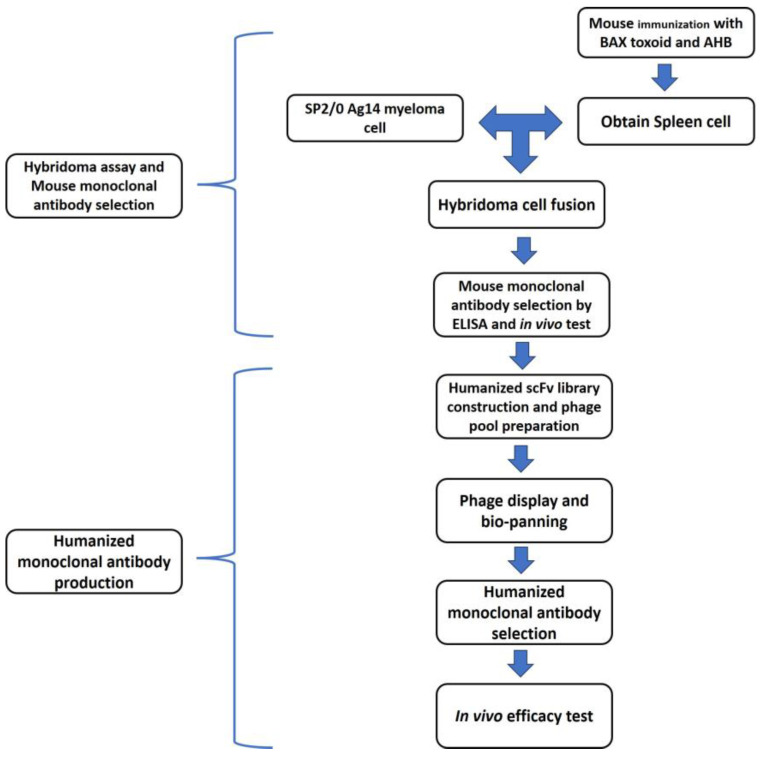
Schematic representation of this study. Splenocytes were prepared from immunized mice. They were fused with a myeloma cell line to obtain hybridoma cells. The murine BoNT/A antibody was selected via ELISA and murine bioassay. The heavy- and light-chain genes of the murine monoclonal antibody were analyzed by overlap PCR. The humanized scFv library was cloned into a phagemid vector. Bio-panning was performed to obtain specific antibodies. The humanized monoclonal antibody was selected via ELISA and murine bioassay. Finally, the selected humanized antibody was tested for therapeutic and prophylactic efficacy.

**Table 1 toxins-17-00138-t001:** Results of in vivo neutralization assays with HZ45.

HZ45 Dose (μg)	BoNT/A Challenge (LD_50_)	No. of Surviving Mice/Total No.
100	100	5/5
50	100	0/5
50	20	5/5
10	20	5/5
5	20	5/5
1	20	0/5

**Table 2 toxins-17-00138-t002:** Protection of mice from BoNT/A intoxication by HZ45.

BoNT/A Challenge (LD_50_)	HZ45		No. of Surviving Mice/Total No.	Survival (%)
	Time of Treatment	Dose (µg)		
20	24 h Pre-administration (Mock IgG treatment)	100	0/6	0
20	24 h Pre-administration	100	6/6	100
100	24 h Pre-administration	100	6/6	100
20	24 h Pre-administration	50	6/6	100
20	24 h Pre-administration	10	3/6	50
20	4 h Post-administration	100	6/6	100
20	8 h Post-administration	100	6/6	100
20	24 h Post-administration	-	Moribund before treatment	0
50	8 h Post-administration	-	Moribund before treatment	0

**Table 3 toxins-17-00138-t003:** Protection of rabbits from BoNT/A intoxication by HZ45.

BoNT/A Challenge (LD_50_) *	HZ45		No. of Surviving Rabbits/Total No.	Survival (%)
	Time of Treatment	Dose (mg)		
10	24 h Pre-administration (Mock IgG treatment)	0	0/4	0
10	24 h Pre-administration	5	4/4	100
10	Co-administration	5	4/4	100
10	4 h Post-administration	5	4/4	100
10	8 h Post-administration	5	4/4	100

* BoNT/A LD_50_ for rabbits: 1.352 ng/kg.

## Data Availability

The original contributions presented in this study are included in the article/Appendix A. Further inquiries can be directed to the corresponding author.

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
