# Peer review of "An Effective Prophylactic and Therapeutic Protection Against Botulinum Type A Intoxication in Mice and Rabbits Using a Humanized Monoclonal Antibody"

_toxins, 2025, doi:10.3390/toxins17030138_

Round 1
Reviewer 1 Report (Previous Reviewer 2)
Comments and Suggestions for Authors
The authors have addressed my comments raised in previous submissions.
Author Response
Review is completed
Reviewer 2 Report (Previous Reviewer 1)
Comments and Suggestions for Authors
This manuscript shows much improvement. The English is much better, and the studies and results have been greatly clarified. I have listed a few minor revisions. The only major issue with the manuscript is the authors’ continuing insistence on emphasizing inhalational botulism as a threat and a naturally occurring cause of botulism. Practically speaking, inhalational botulism is neither, and the threat to infants and through foodborne botulism is a much more relevant issue than theoretical inhalation botulism. I would like to see this clarified and de-emphasized, and inclusion of infant botulism as a natural cause of botulism should be added.
Line 32-33 – You have omitted the natural occurrence of infant botulism, which in many countries is the most common form of botulism seen. Also, you MUST delete inhalational botulism as a naturally occurring form of botulism. There has never been a recorded case of naturally occurring inhalation botulism (or any recorded case of inhalational botulism, for that matter). I have stated this several times. Your citations do not support the statement that inhalational botulism is a natural form of botulism, only that it is theoretically possible based on studies in nonhuman primates.
Line 33-34 –Botulism is not considered to be a medical emergency, it just requires significant clinical care in severe cases. Thus, a large number of severe botulism cases at the same time would be considered a medical emergency. Botulism can manifest as barely noticeable changes in sight or voice up to severe paralysis and difficulty breathing, but prompt effective treatment options have reduced the threat of fatalities to a very few each year.
Line 34-35. The fatal inhalation and ingestion doses are theoretical, based on data in laboratory animals. This sentence needs to begin with “theoretically”.
Line 39 – BoNTs are not considered Category A biological agents because they can cause inhalational botulism, they are considered Category A agents because of their ability to cause widespread harm to people, particularly if intentionally used. Contaminating food or water supplies may well be a more realistic scenario than aerosol release, as demonstrated by the failure of Aum Shinrikyo to intoxicate people in Japan after aerosol release of botulinum toxin. You can simply state that BoNTs are category A select agents. As mentioned above, BoNTs have never actually caused inhalational botulism in humans, so again, you must use the word, “theoretically” or delete the sentence.
Line 40-44 – BoNTs have never successfully been used as bioterrorist agents. The events in Tokyo were an unsuccessful attempt by the domestic terrorist group, Aum Shinrikyo, to use botulinum toxin as an aerosol agent (which failed, telling you something about the use of botulinum toxin as an aerosol agent), and, while Iraq and Russia have stockpiled large quantities of botulinum toxin, there is no evidence that BoNT was used as a bioterrorism agent. This statement is inaccurate and it is not supported by the citation (4). It must be kept in mind that the modeling study cited is strictly theoretical and it was done over 25 years ago.
Line 58 – “the use of single”, not “to use single”.
Line 73 – This needs an introductory sentence, such as “The preparation was subjected to three rounds of panning”.
Line 125 – “anti-cleavage” should be changed to “prevention of cleavage”.
Line 201-210 – This information should more appropriately be within section 4.3. It reads more like methods than discussion. I would place information related to the framework in the methods and materials section and specific results numbers on library diversity in the methods and materials section (or possibly results).
Line 211-215 - This information should more appropriately be within section 2.1. It reads more like results than discussion.
Line 224 – A more appropriate citation here would be Razai, 2005, J Mol Biol 351:158-169, instead of Nowakowski, where anti-BoNT/A antibodies with picomolar to sub-picomolar affinities were constructed (1.71 and 0.554 x 10-12 M-1). This paper also discusses the issue of inaccurate affinities using BIAcore with antibodies having very high affinities, which is helpful to know.
Line 290 – It is essential that the toxin subtype be listed, as antibody affinity and neutralization ability varies greatly against the different BoNT/A subtypes (re: Smith, 2005). The strain used would also be desirable.
Line 303 – The final toxicity of the BoNT/A toxin needs to be stated (e.g., 1.0 x 106 LD50/ml).
Line 315 – Are the authors certain that they dialyzed the toxin in formaldehyde at 35 C for 7 days? This would typically be done at either room or refrigerator temperature. Dialyzing at such a high temperature may have damaged the structure of the toxin. Could the authors provide a reason for dialysis at such a high temperature?
Line 327 – euthanized, not euthanatized.
Line 460-461, 464-465 – It appears that some information needs to be filled in (***).

Author Response
Please see the attachment

Reviewer 3 Report (New Reviewer)
Comments and Suggestions for Authors
The manuscript presents a thorough investigation into the development and evaluation of a humanised monoclonal antibody (HZ45 mAb) for prophylactic and therapeutic protection against botulinum neurotoxin type A (BoNT/A). The study encompasses various experimental methodologies, including hybridoma technology, phage display, in vitro assays, and in vivo murine and rabbit models, to demonstrate the efficacy and binding affinity of HZ45.
Abstract
- The abstract presents key results but does not mention statistical significance or confidence intervals for the survival data. This omission makes it difficult to assess the robustness of the findings.
- While it states that HZ45 neutralises BoNT/A, it does not clarify whether the antibody blocks toxin entry, inhibits intracellular activity, or accelerates clearance. A brief mention of the mechanism would strengthen the summary.
- The abstract states that a 100 mg dose was used in rabbits, but in the results, a 5 mg per rabbit dose is reported. This discrepancy needs clarification to avoid confusion.
Introduction
- The introduction repeats information about botulinum neurotoxin’s toxicity and potential use as a biological weapon. While emphasising its importance is valid, the section could be more concise without losing clarity.
- The introduction acknowledges that a combination of multiple mAbs increases potency and reduces escape mutants but does not convincingly justify why a single mAb approach was chosen in this study.
- Although previous studies on monoclonal antibodies against BoNT/A are cited, the introduction does not critically compare how HZ45 differs or improves upon them.
Results
- The authors mention selecting 12 clones but provide no criteria for exclusion or detailed binding kinetics of non-selected clones. This weakens the rationale for choosing HS45 as the final candidate.
- The binding affinity is presented, but error bars, p-values, or confidence intervals are missing. This weakens the reliability of ELISA data and makes reproducibility uncertain.
- While the protection efficacy is demonstrated in mice and rabbits, no placebo control (e.g., non-neutralising IgG treatment) is explicitly mentioned. The inclusion of such controls would strengthen conclusions.
- The manuscript does not assess whether HZ45 induces immune reactions, cytokine release, or other adverse effects in animals. This is a significant oversight for a therapeutic candidate.
- Some data tables, particularly Table 1 and Table 2, are structured in a way that makes it difficult to interpret survival outcomes. The readability could be improved with clearer column headings.
Discussion
- The discussion heavily references previous monoclonal antibody research, but there is no direct performance comparison (e.g., binding affinity, half-life, efficacy in LD50 protection).
- The study assumes that a single monoclonal antibody can effectively neutralise BoNT/A without considering the potential for mutation-driven resistance. Given that previous studies emphasise the advantage of multi-mAb therapies, this is a major gap.
- The use of New Zealand White rabbits is justified for in vivo testing, but no reasoning is given for why primates or other larger animals were not considered—especially since primates are more physiologically relevant to humans in botulinum studies.
- Since this is a humanised monoclonal antibody, the discussion should consider the potential for human anti-human antibody (HAHA) responses, which could limit its clinical effectiveness.
- The discussion does not outline the next steps needed for clinical translation, such as pharmacokinetics, toxicology studies, or large-scale production feasibility.
Materials and Methods
- The number of animals used per group is not justified statistically. A power calculation should have been provided to demonstrate whether the sample size is adequate for detecting a significant effect.
- The paper describes phage display selection but does not clearly define the criteria used to rank and finalise clones. For instance, did the authors prioritise high-affinity binders or those with the best neutralization efficacy?
- The purification of BoNT/A toxin is described, but details regarding endotoxin contamination removal are missing. This is important because trace endotoxins could affect assay results in animal models.
- The SNAP-25 assay lacks proper controls, such as irrelevant mAb treatment or untreated toxin-only groups, which are essential to confirm that neutralization is specific.
- The manuscript states that "529 clones were screened, and 14 were selected," but no details are provided on how enrichment was assessed over rounds of selection.
Conclusion
- The conclusion states that HZ45 "shows promise as an antitoxin for human use," but no pharmacokinetic or toxicity data is presented to support this claim. A more cautious phrasing would be appropriate.
- Large-scale production of humanised mAbs can face challenges in yield, stability, and cost, but the manuscript does not mention any potential hurdles in HZ45’s development.
- The study does not mention what further FDA or EMA regulatory studies would be required before moving to human clinical trials.
Author Response
Please see the attachment.

Reviewer 4 Report (New Reviewer)
Comments and Suggestions for Authors
Line 93: provide confidence intervals for the KD measurements
Line 251: Remove or fix the sentence: “Initially, therefore, our experiments were performed at half of the NOAEL (no-observed-adverse-effect level).” The authors did not administer the HZ45 to determine the adverse effect level of the antibody.
The authors should comment on the amount of HZ45 that would have to be administered to a patient to show equivalence to the potency of BAT. Is this a reasonable dose?
Table 3: what was the clinical status of the rabbits 4 and 8 hr post toxin administration?
Figure 4. What is ABN901? Was an experiment done where BoNT A administered before HZ45?
Author Response
Please see the attachment.

This manuscript is a resubmission of an earlier submission. The following is a list of the peer review reports and author responses from that submission.
Round 1
Reviewer 1 Report
Comments and Suggestions for Authors
This manuscript describes the development and characterization of a humanized monoclonal that is capable of neutralizing botulinum neurotoxin type A (BoNT/A). There are a number of issues with this manuscript, foremost of which is that the English is not adequate for publication. As an example, “The mAbs that are well defined specificity could lessen its toxicity relative immunized serum and human or humanized mAbs can treat botulism without side effect compare to equine antitoxin” is incomprehensible. A complete re-write needs to be accomplished before a serious review can be done.
Secondly, the results and materials and methods sections have some serious issues.
Section 2.1/4.4. There are serious problems with the materials and methods (M&M) section versus the results section here. There is an explanation of the method for generating hybridomas, but there is no information in the results section on how many hybridomas were produced and tested for generation of anti-BoNT/A antibodies. The results describe testing of monoclonal antibody in a mouse bioassay, but they do no describe in either M&M or results how they generated and purified soluble antibodies prior to testing, nor did they indicate how many assays were done, or what the results of these assays were. After reading the entire paper, I seem to understand that the neutralizing antibody characterized here came form the phage library, not the hybridomas, so I am left wondering why the authors chose to provide this information, if it is irrelevant. In addition, this section is titled, “Hybridoma and phage display assay of selection of neutralizing antibodies”, but there is no information whatsoever relating to a phage display assay. The authors need to remove the hybridoma sections if the antibody in question (HZ45) was produced using phage display, or remove the phage display sections if HZ45 was produced from a hybridoma.
Section 2.2/4.5/4.6. There are also serious problems with these two sections. The M&M sections are sparse on details and do not cite a single reference. If these were PCR assays that were unique to the authors, then primers/conditions, etc. need to be added. If they were based on the work of another laboratory, that needs to be referenced. This is basic to scientific writing, and the authors should know that. With the results section, it appears that following an initial biopanning, a single E. coli colony (that supposedly contained a single phage particle) was used to create a phage pool that was then subcloned into helper phage and this “pool” produced 529 clones, 45 of which were selected for screening (no mention of how this was accomplished)? All of the “subclones” would be identical if you started with a single colony, so why did the authors do this exercise? And why did they do additional panning of a single clone? This section is mis-titled as “isolation of humanized BoNT/A monoclonal antibody”, but it describes the isolation of a phage clone producing mouse scFv, not humanized monoclonal antibody. Figure 1 is supposedly the sequence of the mouse scFv. Related to this, it is stated in the text that “As a result, it was confirmed that they had the same sequence”. Since the authors state “a HZ45 clone having the highest affinity and neutralization ability”, my question is, who is “they”?
Section 4.7. There is no real information in the M&M section on how the humanized HZ45 antibodies were constructed. Herceptin is a protein antibody, not a genetic sequence. Herceptin cannot be used as a framework. The Herceptin gene sequence might have been used as a framework. You absolutely need to reference where you found this sequence and what publication lead you to use this sequence as a framework. This lack of referencing is deplorable. In addition, what method of “grafting” did you use? Explain in detail or reference.
Section 2.3/4.8. In the M&M section, there is mention of codon-optimization, but no explanation of why that was needed, or what actual changes were made and why. No citations, no details. There is also no reference concerning the expression vector. If this was a commercial vector, then the section about “choosing” Herceptin seems a bit overstated, since the commercial suppliers were the ones who made the choice.
Section 2.4/4.10. This is not an in vitro neutralizing assay. It is a cell-based assay. There is an enormous difference. An in vitro SNAP-25 assay determines if an antibody is able to directly interfere with interaction of SNAP-25 or a SNAP-25 peptide fragment to the BoNT/A enzymatic site. A cell-based assay determines whether the toxin is able to bind to and enter cells in oreder to prevent entry and cleavage of BoNT/A inside cells. Thus, antibodies that prevent binding and internalization of toxin prevent SNAP-25 cleavage in a cell-based assay, while antibodies that block the enzymatic site prevent SNAP-25 cleavage in an in vitro assay. If this were an in vitro assay, HZ45 would fail to protect. There is also a huge problem with section 4.10. Neuro-2A cells were used in this assay, but the cited reference for the procedure used primary chick embryo cells, not Neuro-2A cells. In addition, the authors treated the cells with toxin/antibody for 2 days. Other authors (Blum, 2014; Fan, 2015) that did similar testing using Neuro-2A cells treated them for 30 minutes-2 hours. The authors need to explain why they chose a 2 day incubation, and how this exceptionally long incubation period might have affected the results.
Section 2.5/4.1/4.11. In vivo mouse bioassays were done using a premixed combination of antibody and toxin followed by i.p. injection. Typically, these assays are done using no more than 50 ug of antibody versus toxin challenges starting at 10 or 20 LD50. The authors used up to 1 mg of antibody per mouse, which is 20 times more antibody, and which represents an almost 8,000-fold excess of antibody versus toxin, according to the LD50 of the BoNT/A that was used in the studies (Yu et al, 2028). Since binding of single antibodies to toxin does not allow for rapid clearance of antibody-toxin immune complexes through the liver in the same way that multiple (oligoclonal) antibody does (Al-Saleem et al, 2111. J Pharmacol Exp Therapeut 338:3043-3054), protection with single antibodies relies on a combination of antibody affinity and antibody excess to prevent binding and entry of sufficient toxin to cause death before it can be naturally cleared. This is a matter of simple pharmacokinetics. Most neutralizing anti-BoNT antibodies have subnanomolar affinities, so that antibody amount is a factor. The greater the antibody excess, the longer it takes to develop and fatal level of toxin within target neuron cells, and the greater the neutralizing ability of the antibody within the timinig framework of the study, so the use of unusually high amounts of antibody explains the higher protection seen here. These results also prompted an investigation of the toxin that was used in these experiments. The toxin subtype and strain used for its production are not stated, but it is assumed that the toxin subtype is BoNT/A1. Each of the eight BoNT/A toxin subtypes has its own potency characteristics, and BoNT/A1 is nown to be highly potent, with typical mouse LD50 of 2-5 x 107/mg. The toxicity of this BoNT/A is approximately one log lower (7.7 x 106 LD50/mg) than that of standard purified BoNT/A1 preparations, which raises questions about the toxin subtype/strain that was used to produce this toxin, and its quality. Can the authors elaborate on the toxin subtype and explain why this toxin has such a low potency?
Section 2.6. Mice challenged with 20 LD50 BoNT/A do not typically appear moribund 4 hours after challenge. Are the authors sure about their toxin doses? Charts showing the timing of the deaths of the animals in the results sections of these studies would be also helpful. Were all the deaths early, or spaced out at different intervals?
Section 2.7/4.11. I am assuming that with the rabbits the toxin challenges were in mouse i.p. LD50. What is the relationship between mouse and rabbit LD50? How many mouse LD50 = 1 rabbit LD50? The rabbit studies were poorly designed and are unrealistic when assessing human therapeutic use. The rabbits were dosed at 100 mg per animal (~40 mg/kg) with antibody:toxin excesses ranging from 8 to 40 million-fold. An equivalent human dose would be a whooping 2.8 grams per person to protect against a relatively small toxin dose. Also, there is no protective endpoint reached, so that we have no idea if 100 LD50 is the protective limit or not.
The authors state that they have shown potent prophylactic and therapeutic protection against BoNT/A intoxication in mice and rabbits using a single humanized monoclonal antibody, but the actual data does not support this and indicates what we already know - there is a limited therapeutic window for treatment of BoNT intoxication, and success using a single or multiple antibody preparation is ensured only by treating with a large excess of antibody before symptoms of intoxication occur. This has been known and demonstrated using other monoclonal and polyclonal antibody preparations many times. The authors should present this manuscript as describing an effective monoclonal neutralizing antibody with prophylactic and therapeutic potential, not as an antibody showing potent prophylactic and therapeutic protection, because the data to support this is simply not shown here.
As stated in the introduction, polyclonal or oligoclonal antibody make vastly superior prophylactic and therapeutic agents against botulinum neurotoxins than single antibodies. Triple antibody combinations containing less than 50 ug of total antibody are known to protect against BoNT/A challenges of up 40,000 LD50, which is much greater than single antibody protection. While the quality control of such products is more complicated than with single antibodies, single antibody preparations will never be viable therapeutic treatments for botulism due to their lower relative affinities and their inability to rapidly clear toxin from the circulation. However, this antibody would be a promising candidate as a therapeutic in combination with other antibodies.
A more thorough review of this manuscript cannot be undertaken until after the English is corrected.
Comments on the Quality of English LanguageThe English is not up to the standards required for submission in this journal.
Reviewer 2 Report
Comments and Suggestions for Authors
The authors have produced an interesting and novel humanized monoclonal antibody against botulinium type A toxin which has therapeutic and prophylactic potential. However the manuscript is very disjointed and in parts quite repetitive and is thus in need of major reorganization and revision.
Major comments
Section 2 should comprise Material and methods and be organized in a logical and progressive order, much of which appears in their section 4. Hence the materials and methods section (2) should start with purification of the toxin followed by antigen preparation, immunization, hybridoma fusion and screening, library construction etc so that the reader can follow the methodological sequence. Sections 2.3 and 4.9 for example on SPR methodology should be combined and condensed. Similarly for animal studies section 4.11 and sections 2.6 and 2.7 should be rationalized and condensed. The authors need to be aware that the methods section should not contain results and that the results section should minimize any method description.
The discussion section needs to be rewritten and condensed to avoid any repetition of the introduction.
Other comments
Generally abbreviations should be explained when used for the first time e.g. scFV, CDR
Introduction line 44 8.6 billion USD?
There needs to be a legend for figure 1.
What was the source of anti-SNAP antibody and the HCBD for the ELISA screening.
Line 343 page elution is with triethylamine
Comments on the Quality of English Language
Apart from major reorganization of the manuscript the English needs editing as well.
Reviewer 3 Report
Comments and Suggestions for Authors
I have reviewed the manuscript entitled “A Potent Prophylactic and Therapeutic Protection against Botulinum type A Intoxication in Mice and Rabbits by Humanized Monoclonal Antibody” and found interesting although I have some minor query before acceptance.
1. Author should add the schematic representation of study for better understanding of audience.
2. Include space between numbers and units, spell out units of time, and clarify the sentence structure in "100mg of HZ45 could treat NZW within 8hour after exposure of 20LD50 botulinum"
3. Use the correct unit symbol (µg instead of ug) for micrograms in a sentence “A 70 kg person can be killed by inhaling 0.7 to 0.9 ug, or by ingesting 70 ug" should be "A 70 kg person can be killed by inhaling 0.7 to 0.9 µg, or by ingesting 70 µg."
4. Add a space between "100" and "mg" and "4" and "hour." In this sentence "These results indicate that 100mg of HZ45 could treat NZW within 4hour after exposure of 100LD50 BoNT/A."
5. Ensure consistent unit usage (µg instead of microgram). "One microgram of HZ45 did not protect mice challenged with BoNT/A 20 LD50."
6. Correct the phrasing for clarity. "Cleavage of SNAP-25 was decreased with the dependent of antibody concentration."
7. Clarify sentence structure for better readability in a sentence “Since that domain attributes to cellular recognition of the toxin for transmembrane transport, neutralizing ability of this antibody might come from inhibiting cellular uptake of the toxin."
8. Correct "0.042M" to proper unit. "The resulting HZ45 bound to the HCBD with an apparent Kd of 0.042M, an affinity 10-fold higher (lower Kd) than that of BoNT/A."
9. Correct spelling to "hybridoma." "Adekar et al developed two IgG, 6A and 4LCA, that were generated by hybrioma technology from humans immunized against pentavalent botulinum toxoid."
10. Correct "was death" to "died." "PBS control group was death at 24-28 hour against 20LD50 of BoNT/A."

Round 2
Reviewer 1 Report
Comments and Suggestions for Authors
see attached file.

The quality of English here was improved, especially pertaining to spelling and grammar, but it was still poor enough that I was unable to clearly understand the meaning of some of the statements.
Reviewer 2 Report
Comments and Suggestions for Authors
Although a few points were addressed my major criticism about the organization of the manuscript still exists. Section 2 must be Materials and methods. It should start with 2.1 Botulinum toxin purification and then progress right through to 2.11 Animal Studies. Hence the authors must relabel all this section.
Section 3 must be 3. Results and it hence must also be relabeled.
The original Figure 1 is now Figure 2 in revision 1 which also needs correcting in the text. The sensorgrams are now Figure 3 which needs correcting in the legend and in the text. Likewise for Figure 4 which now shows "Protection of protein by the HZ45 antibody" and also needs correction in the text.
I think the author(s) or editor could further refine the manuscript so that methods appear in the Materials and methods section rather than in the results.
Comments on the Quality of English LanguageSome improvement and numbering adjustments are needed in the manuscript.
